# A comparison of the attractiveness of flowering plant blossoms versus attractive targeted sugar baits (ATSBs) in western Kenya

**Nick Yalla[1]\*, Brian Polo[1], Daniel P. McDermott[2], Jackline Kosgei[1], Seline Omondi[1], Silas Agumba[1], Vincent Moshi[1], Bernard Abong'o[1], John E. Gimnig[3], Angela F. Harris[4], Julian Entwistle[4], Peter R. Long[5], Eric Ochomo[1]\***

**1** Entomology Department, Kenya Medical Research Institute, Centre for Global Health Research, Kisumu, Kenya, **2** Department of Vector Biology, Liverpool School of Tropical Medicine, Pembroke Place, Liverpool, United Kingdom, **3** Division of Parasitic Diseases and Malaria, Centre for Global Health, Centers for Disease Control and Prevention, Atlanta, GA, United States of America, **4** Innovative Vector Control Consortium, Liverpool, United Kingdom, **5** Department of Biological and Medical Sciences, Oxford Brookes University, Gipsy Lane, Oxford, United Kingdom

\* nickyalla60@gmail.com (NY); ericochomo@yahoo.com (EO)

## Abstract

Attractive Targeted Sugar Baits (ATSB) have been demonstrated to result in significant reductions in malaria vector numbers in areas of scarce vegetation cover such as in Mali and Israel, but it is not clear whether such an effect can be replicated in environments where mosquitoes have a wide range of options for sugar resources. The current study evaluated the attractiveness of the predominant flowering plants of Asembo Siaya County, western Kenya in comparison to an ATSB developed by Westham Co. Sixteen of the most common flowering plants in the study area were selected and evaluated for relative attractiveness to malaria vectors in semi-field structures. Six of the most attractive flowers were compared to determine the most attractive to local *Anopheles* mosquitoes. The most attractive plant was then compared to different versions of ATSB. In total, 56,600 *Anopheles* mosquitoes were released in the semi-field structures. From these, 5150 mosquitoes (2621 males and 2529 females) of *An. arabiensis*, *An. funestus* and *An. gambiae* were recaptured on the attractancy traps. *Mangifera indica* was the most attractive sugar source for all three species while *Hyptis suaveolens* and *Tephrosia vogelii* were the least attractive plants to the mosquitoes. Overall, ATSB version 1.2 was significantly more attractive compared to both ATSB version 1.1 and *Mangifera indica*. Mosquitoes were differentially attracted to various natural plants in western Kenya and ATSB. The observation that ATSB v1.2 was more attractive to local Anopheles mosquitoes than the most attractive natural sugar source indicates that this product may be able to compete with natural sugar sources in western Kenya and suggests this product may have the potential to impact mosquito populations in the field.

**Data Availability Statement:** All relevant data are within the paper and Supporting Information files.

**Funding:** This study was funded by the Innovative Vector Control Consortium (IVCC), UK, which is funded by grants from the Bill and Melinda Gates Foundation and other funding partners. AF and JE are consultants with IVCC and offered technical support in the design of this trial but had no role in the data collection and analysis of this manuscript. They reviewed and approved this manuscript.

**Competing interests:** The authors have declared that no competing interests exist.

## Background

Current vector control tools, such as long lasting insecticidal nets (LLINs) and indoor residual spraying (IRS) have contributed substantially to the reduction in the global malaria burden since the early 2000s [1,2]. These strategies and others currently under evaluation target known mosquito vulnerabilities such as indoor, late night blood feeding, and resting behaviors to control them [3]. However, other life history traits such as sugar feeding have remained largely understudied although sugar feeding has recently been identified as a potential vulnerability that could be exploited for vector control [4–7].

Mosquitoes of both sexes require sugar for energetic needs and survival [8,9] and their main natural sources are flowering plant nectaries, ripe-rotting fruits, or honeydew [10]. Mosquitoes are thought to identify sugar meals by volatiles emitted by floral blossoms which they eventually cue in to [11]. Attractive targeted sugar baits (ATSB) contain an attractant, sugar source as a feeding stimulant mixed with an oral toxicant to kill mosquitoes upon ingestion by exploiting the sugar feeding behavior of mosquitoes to reduce their populations [12]. For ATSBs to optimally impact mosquito populations, they would have to compete favorably with the available natural sugar resources. Several preferred sugar sources for mosquitoes have been described in the literature [10,13–15], some of which have been exploited for vector control and surveillance [16,17].

ATSB applications have been demonstrated to effectively reduce mosquito densities. In Mali, a 90% reduction in the density of *An. gambiae sl.* and a reduction in the longevity of female mosquitoes were observed following ATSB spot spraying of non-flowering vegetation around artificial ponds [4]. Similarly, the density of *An. sergentii*, one of the principal potential malaria vector in Israel, was reduced by >95% when the sugar-rich vegetation around oases was sprayed with ATSB solutions [5,7] resulting in significantly lowered biting pressures. Most recently in Mali, ATSB stations markedly reduced the density of *An. gambiae*, including a 90% reduction of older females who had survived three or more gonotrophic cycles and were therefore potentially infectious was reduced by 90% [18]. Importantly, ATSBs can be used to target a wide range of mosquito vector species and have been shown to be efficacious against resistant mosquito populations of *Aedes aegypti*, and *An. arabiensis* in semi-field and laboratory evaluations [19,20]. ATSBs are therefore a potentially effective complementary addition to the existing malaria vector control tools.

To date, most successful field evaluations of ATSB have been conducted in arid/semi-arid environments where the baits are likely to face little competition due to scarcity of natural sugar sources such as Mali [4,18] and Israel [5,7]. The efficacy of ATSB in more vegetated environments where mosquitoes may have a wide range of options for sugar sources remains largely unexplored. This study, therefore, aimed to compare the relative attractancy of mosquitoes to different versions of an ATSB manufactured by the Westham Co. against a variety of flower blossoms of plants found in western Kenya. These findings will be key to identifying the potentially competitive plants and estimating how they may affect the deployment of ATSBs during the epidemiological evaluations that are currently taking place in three countries in Africa including Kenya [21] and eventually during public health campaigns.

## Methods

### Study site

The attractancy experiments were conducted in three semi-field structures measuring 18m x 9m x 3m within the Kenya Medical Research Institute (KEMRI) located at the Kisian research station in Kisumu, Kenya, as described in [22]. Prior to the semi-field experiments, a team of

entomologists and botanists conducted a botanical survey in Asembo (0.183694˚S, 34.383694˚E) Siaya County to characterize the common flora of the area. During the survey, the coverage of all flowering plants around the peridomestic spaces was estimated across ten clusters between May-August 2020. The clusters were made up of several adjacent villages with 100–400 households [21]. The most common flowering plants that were potential sugar sources for mosquitoes were identified by the botanist with the aid of a mobile app (PlantSnap® app, Eden Tech Labs; Sofia, Sofia-City; Bulgaria). Flowers tested in attractancy experiments were the most common flowering plant species collected fresh daily from Asembo where a large-scale, cluster-randomized trial of ATSBs is being conducted. The area receives seasonal rainfall with the heaviest long rains between March and May and short rains between October and December. The average annual rainfall is between 1000–1250 mm and the daily maximum temperature ranges between 17-35˚C. It lies at a mean altitude of 1070 m above sea level. The area is malaria endemic with transmission occurring throughout the year. Asembo forms part of the KEMRI health and demographic surveillance system (HDSS) continuous malaria surveillance site where over 223,000 individuals have been monitored for more than a decade [23–25].

## Collection and maintenance of mosquitoes for bioassays

*Anopheles gambiae* complex mosquitoes were collected as larvae, *An. arabiensis* from Ahero (0.16˚S, 35.19˚E) in Kisumu County and *An. gambiae* from Bumula (0.5742˚N, 34.4420˚E) in Bungoma County using larval dipping. *Anopheles funestus* were collected as adults from Alego-Usonga (0.0606193˚N, 34.2859671˚E) in Siaya County. The field-collected larvae were raised under laboratory conditions with temperature maintained between 27–30˚C and relative humidity 80 ±10%. The larvae were fed on TetraMin fish food up to the pupal stage after which they were transferred to plastic cups and placed in 30x30x30cm cages. Adults were maintained on a 10% sugar solution soaked in cotton wool until they were to 3–5 days post-emergence for bioassays. Adult *An. funestus* were collected from the field using a combination of mechanical and mouth aspiration and transported back to the insectary in Kisumu, sorted by abdominal status so that only fed and gravid *An. funestus* were maintained in separate cages. The rest were discarded. The sorted cages were maintained on 10% sugar for at least 48 hours after which a laying pad was introduced into the cages with gravid mosquitoes. Eggs were collected the next day and hatched overnight. *Anopheles funestus* larvae were reared using the same conditions for *An. gambiae* described above. Previous studies have shown that *An. gambiae* are the predominant species in Bungoma, *An. arabiensis* is the main species in Ahero while *An. funestus* is the major malaria vector in Siaya [26–28]. Thus, mosquito identification was done morphologically. However, PCR results conducted as part of a separate study confirmed these species distributions Kosgei *et al.*, [unpublished]".

## Field experiments

Initial experiments were conducted in an open field in the study area where the most common flowering plants were tested for attraction to malaria mosquitoes. The field site was centrally located in the community and served as a common grazing ground for livestock. This site was chosen for the experiments due to its proximity to mosquito larval sites and for the security of traps which was provided by the community members. Sixteen flowering plants and water as negative control were tested. A total of 17 glue-netted traps were positioned 10 meters apart in a straight line in the field with each enclosing either a test flower or water as a negative control. The glue-netted trapping method was adapted from Müller GC et al. [10] to test fruits, seedpods, and flowers for attraction to malaria mosquitoes. In brief, a black plastic netting with a

mesh-size of 0.5cm x 0.5cm was cut into 70cm x 70cm pieces and folded into cylinders then fixed with plastic seals to form the traps. The top part of the cylinders was enclosed with the same plastic netting. The netted traps were supported with two 20 cm sticks firmly fixed into the ground on either side to prevent them from toppling during strong winds. To the outer surfaces of the cylinder netting, an even layer of a non-toxic, odorless Tangle foot® Tangle-Trap® Sticky Coating (The Scott Company, Marysville, OH, USA) was applied to trap any mosquitoes which cued into the floral scents to seek nectar. The glue was applied daily prior to the experiment to enhance its stickiness. The traps bearing flowering plants were rotated daily in the field to eliminate the effect of positioning bias. However, only 132 culicine mosquitoes were caught on the glue-netted traps in a period of 20 days but with no anopheline species, so the glue-netted trap experiment was transferred to the semi-field structures.

### Semi-field experiments

In an 18m x 9m section of the semi-field structures, glue netted traps similar to those previously used in the field experiment (Fig 1) were used. The traps were placed at each of the four corners of the release area of the semi-field structure at a distance of 10m x 8m apart. Empty 500ml plastic jars were buried in the ground and filled with tap water. Wearing industrial hand gloves, fresh flowers of the flowering plants previously identified and tested in the ATSB study area were cut with a machete and about 0.5 kg (40cm-60cm) were placed in the cylinder netting each evening of the experiment with their stems inside the 500ml plastic jar with tap water to keep them fresh overnight. One netted trap with only tap water in the 500ml plastic jar was included in each semi-field structure as a negative control so that a total of 3 different flowering plants or ATSBs were tested each night.

Sixteen flowering plants in the study area were tested for attraction to *An. arabiensis* and *An. gambiae* mosquitoes in experiment 1. These plants were: *Parthenium hysterophorous* (Asteraceae), *Mangifera indica* (Anacardiaceae), *Tithonia diversifolia* (Asteraceae), *Leonotis leonurus* (Lamiaceae), *Hyptis suaveolens* (Lamiaceae), *Markhamia lutea* (Bignonaceae), *Senna siamea* (Fabaceae), *Bougainvillea glabra* (Nyctiginaceae), *Ocimum gratissimum* (Lamiaceae), *Lantana camara* (Verbenaceae), *Senna bicupsularis* (Fabaceae), *Crotalaria pallida* (Fabaceae), *Senna dydimobotrya* (Fabaceae), *Tephrosia vogelii* (Fabaceae), *Caranthus roseous* (Apocynaceae) and *Rumex conglomeratus* (Polygonaceae). It was not possible to raise sufficient numbers of *An. funestus* for inclusion in the first semifield experiment comparing the relative attractiveness of these sixteen floral species. Each night, test flowers were randomized for testing in each of the three semi-field structures. At least three randomly selected flowers and a water control were tested in each of the three semi-field structures with both *An. arabiensis* and *An. gambiae*. The test flowers were replicated twice per species and thereafter, the 10 least attractive flowering plants were excluded from subsequent studies.

The six remaining flowering plants that had attracted the most mosquitoes were selected for further tests against different versions of ATSB developed by Westham Co. (Hod-Hasharon, Israel) in the second stage of the experiments. The six most attractive flowers were randomized for the 3 semi-field structures on each experiment day. In each of the 3 semi-field structures, one ATSB version 1.1.1 (also referred to as v1.1) was compared to two different flowering plant species, and water as a negative control. The test materials were rotated each night within the semi-field structures to avoid positional bias. The test materials were replicated four times at this stage with each of the 3 species of mosquitoes. The most attractive flower out of the six was selected and further compared with two versions of ATSBs (versions 1.1 and 1.1.2) and water control in the third stage of the experiments. Nine replicates of these test materials were evaluated with each of the 3 mosquito species. Lastly, in experiment 4,

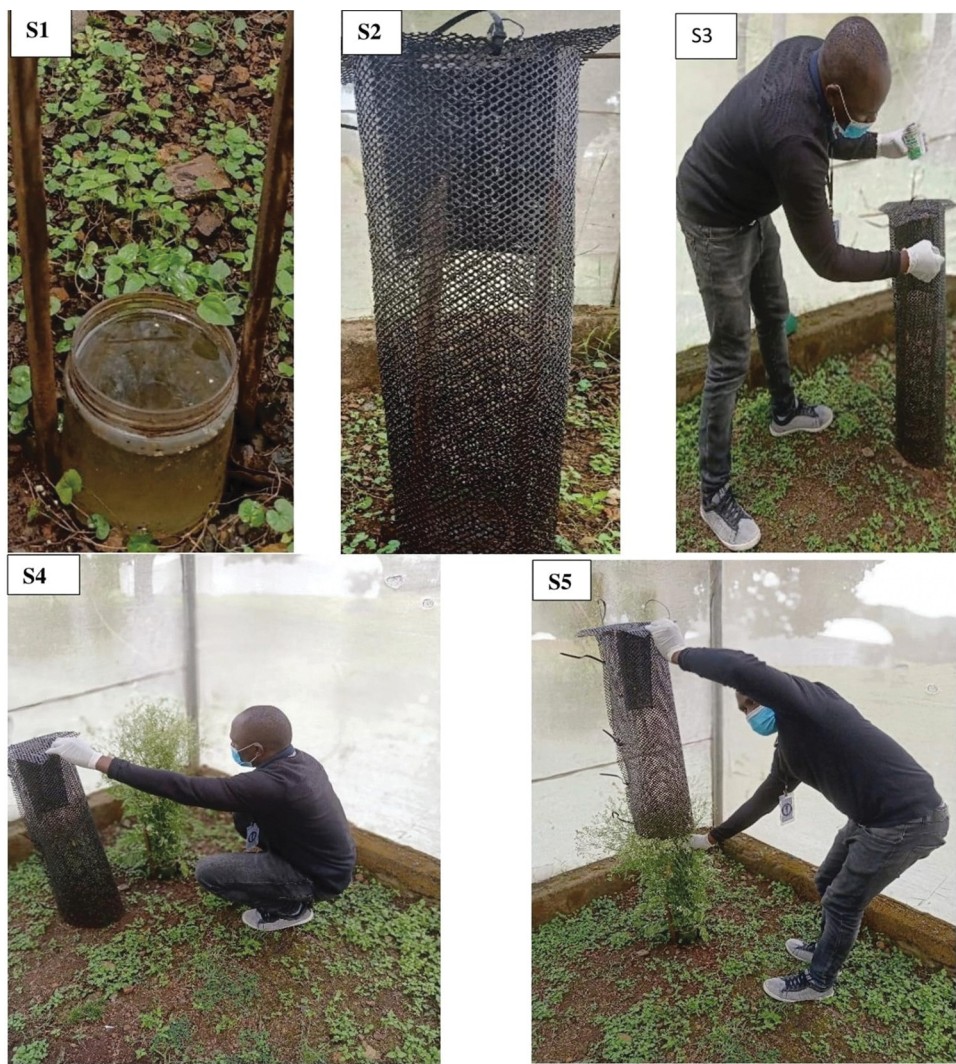

**Fig 1. S1**; a 500ml plastic jar with tap water for keeping the flowering plants fresh. **S2**; Black glue-netted trap and **S3**; shows the application of tangle foot glue on the netted tap using an 8cm handy brush applicator. **S4 & S5**; Fixing flowering plant in glue netted trap within the semi-field structure. On each evening of the experiment, 600 field collected *An. gambiae*, *An. arabiensis* or 200 F1 progeny of *An. funestus* adult mosquitoes with equal numbers of males and females were released at the center of the semifield structures between 1700–1800 HRS and left overnight. All mosquitoes were 3–5 days old and had been starved for 24 hours. The next morning, between 0600-0700HRS, all mosquitoes trapped with the tangle foot glue were picked with a pair of forceps from the glue-netted traps, counted, sexed then transferred to 1.5 ml Eppendorf tubes, and subsequently desiccated in silica gel for archiving (VWR International bvba Geldenaaksebaan 464-B-300I, Leuven, Belgium). In between the experiments, a time gap of at least two days was left for the semi-fields to clear of any remaining mosquitoes from the previous release by natural mortality.

ATSB v1.2 was evaluated against ATSB v1.1, the most attractive floral blossom out of the 6 flowers, and water as the negative control. In this stage of the experiment, 3 replicates of test materials were conducted with each of the 3 mosquito species. During these replications, the two ATSB versions, water control, and the most attractive flowering plant were rotated each night within the semifield structures to avoid positional bias. The ATSB versions tested in this study were versions 1.1, 1.1.2, and 1.2. Version 1.2 is currently under epidemiological trial in Kenya, Zambia, and Mali. All three ATSB versions were similar in bait formulation although

v1.1 and v1.1.2 were manually produced, with V1.1.2 incorporating a rainproof layer to prevent leaking, particularly when rained on. In contrast, v1.2 was machine produced with improved sealing between the tray and bait station membrane. The two previous versions 1.1 and 1.1.2 despite having similarity in bait formulation to v1.2, were key to the improved features of the final version used in the main trials. These ATSBs contained a fruit syrup as an attractant, dinotefuran as the active ingredient which is an oral neonicotinoid insecticide, sugar as a feeding stimulant and Bittrex added to deter human consumption [21]. The dinotefuran insecticide has been indicated to be of low toxicity to humans and has a negligible effect on non-target organisms [29]. In this experiment, the baits were placed inside the sticky netted-traps hence mosquitoes were not expected to access or be able to feed on them.

### Data analysis

Raw data was entered into MS excel workbook then transferred to R (version 3.6.3) for analysis. The mean proportion of mosquitoes recaptured was computed to evaluate the relative attraction of the 16 most dominant floral blossoms and the 6 most attractive flowers for experiments 1 & 2 respectively. For the comparison of the ATSB variants with the most attractive plant identified in experiment 1&2 and the water control, a binomial general linear mixed model (GLMM) was used with a fixed effect for the test material and a random effect for date of experiment and the screen house for each of the 3 species using the lme4 package. The significance of the test material variable was assessed using a log-likelihood ratio test against a null model using the lmtest package. Post-hoc multiple pairwise comparisons were carried out using a Tukey's all-pair comparison using the ghlt package with a Holm-Bonferroni correction to account for multiple testing.

### Ethical consideration

Attractancy experiment procedures were reviewed by the Kenya Medical Research Institute Scientific and Ethical Review Unit (SERU, Approval number: 3613). The study was also approved by the Liverpool School of Tropical Medicine Research Ethics Committee (Approval number LSTM 18–015) and the US Centers for Disease Control and Prevention (Approval number 7112) based on a reliance agreement with KEMRI SERU. All floral materials used in the experiments were picked from non-endangered plant species and with verbal consent from the local community. The individual whose image appears in this manuscript has given written informed consent to publish these case details.

## Results

### Field experiment

After a rigorous 20 days of field experiment within the study site, negligible collections of Anopheles mosquitoes and a total of 132 culicine mosquitoes (105 female and 27 males) were recaptured on glue-netted traps set. Thus, the experiment was shifted to semi-field structures where conducted artificial release of a known number, species and sex of mosquitoes.

   **Semi-field experiment.**   *Number of Mosquitoes by species released and recaptured in the 4 sets of experiments.* A total of 56,600 *Anopheles* mosquitoes were released in the 3 semi-field structures consisting of 28,300 males and 28,300 females in the whole experiment season. From this, a total of 2621 males and 2529 female mosquitoes were recaptured from the traps surrounding the test flowers and the different ATSB versions (Table 1).

**Table 1. The total number of mosquitoes of different species released and recaptured in the 4 experiments.**

| Experiment | Mosquito Species | Total No. Released | | Total No. Recaptured | |
|---|---|---|---|---|---|
| | | Males | Females | Males | Females |
| 1 | An. arabiensis | 3600 | 3600 | 210 | 179 |
| | An. gambiae | 3600 | 3600 | 195 | 341 |
| 2 | An. gambiae | 3200 | 3200 | 206 | 231 |
| | An. arabiensis | 3200 | 3200 | 227 | 164 |
| | An. funestus | 1200 | 1200 | 186 | 258 |
| 3 | An. funestus | 1200 | 1200 | 189 | 170 |
| | An. gambiae | 2700 | 2700 | 261 | 143 |
| | An. arabiensis | 2700 | 2700 | 177 | 149 |
| 4 | An. arabiensis | 2700 | 2700 | 302 | 259 |
| | An. funestus | 1500 | 1500 | 264 | 261 |
| | An. gambiae | 2700 | 2700 | 404 | 374 |

### Attractiveness test of *An. gambiae* and *An. arabiensis* to flowering plant blossoms

In the first experiment comparing the attractiveness of the 16 most common flowering plants to *An. gambiae* and *An. arabiensis*, the six floral species that had the highest recapture rates across both mosquito species and sex were *M. indica*, *C. pallida*, *P. hysterophorous*, *M. lutea*, *L. camara*, and *T. diversifolia*. Floral blossoms of *T. vogelii* and *H. suaveolens* were consistently less attractive to both *An. arabiensis* and *An. gambiae*. The mosquito recapture rates were much higher in *An. arabiensis* than *An. gambiae* but the overall relationship to the dominant floral species was similar (Fig 2). These six flowers with the highest recapture rates were further tested against ATSB v1.1 in the next phase of the experiment.

### Selection of the most attractive flowers for further testing

The second experiments evaluating the six most attractive flower species in comparison to ATSB v1.1, and water control found that *Mangifera indica* had the highest mean percent recapture rates and was the most preferred flowering plant across both sexes of the three mosquito species (Figs 3 and 4). The most attractive flower, *Mangifera indica* was further tested against ATSB v1.1 and v1.1.2 in the third stage of experiments.

### Comparison of *Mangifera indica* flower to the ATSB v1.1 and 1.1.2

In experiment 3, the inclusion of the test material as a variable provided a significant improvement in model fit for *An. gambiae* ($\chi^2$ (3) = 58.37, $p$ = <0.001), *An. funestus* ($\chi^2$ (3) = 54.32, $p$ = <0.001), and *An. arabiensis* ($\chi^2$ (3) = 28.68, $p$ = <0.001). In the subsequent pairwise comparisons, ATSB 1.1.2, tested against *An. arabiensis* ($\mu$ = 1.96%), was the only case where there was no significant difference compared to the water control ($\mu$ = 1.6%; $Z$ = 1.61, $p$ = 0.37) with ATSB 1.1 significantly outperforming it ($\mu$ = 2.95%; $Z$ = -3.28, $p$ = 0.006). The opposite relationship was found in *An. funestus* with ATSB 1.1.2 ($\mu$ = 7.94%) having a greater recapture rate than both ATSB 1.1 ($\mu$ = 6.56%; $Z$ = 1.61, $p$ = 0.368) and *Mangifera indica* ($\mu$ = 5.44%; $Z$ = 2.99, p = 0.014). While there was a significant increase in attraction of *An. funestus* to ATSB 1.1.2 than the dominant sugar source *Mangifera indica*, across the three species the absolute difference in recapture rates was small overall and would not indicate a substantial increase in attraction to the ATSBs compared to an available natural source but rather show that in its previous iterations, the ATSBs were as competitive in attracting malaria mosquitoes

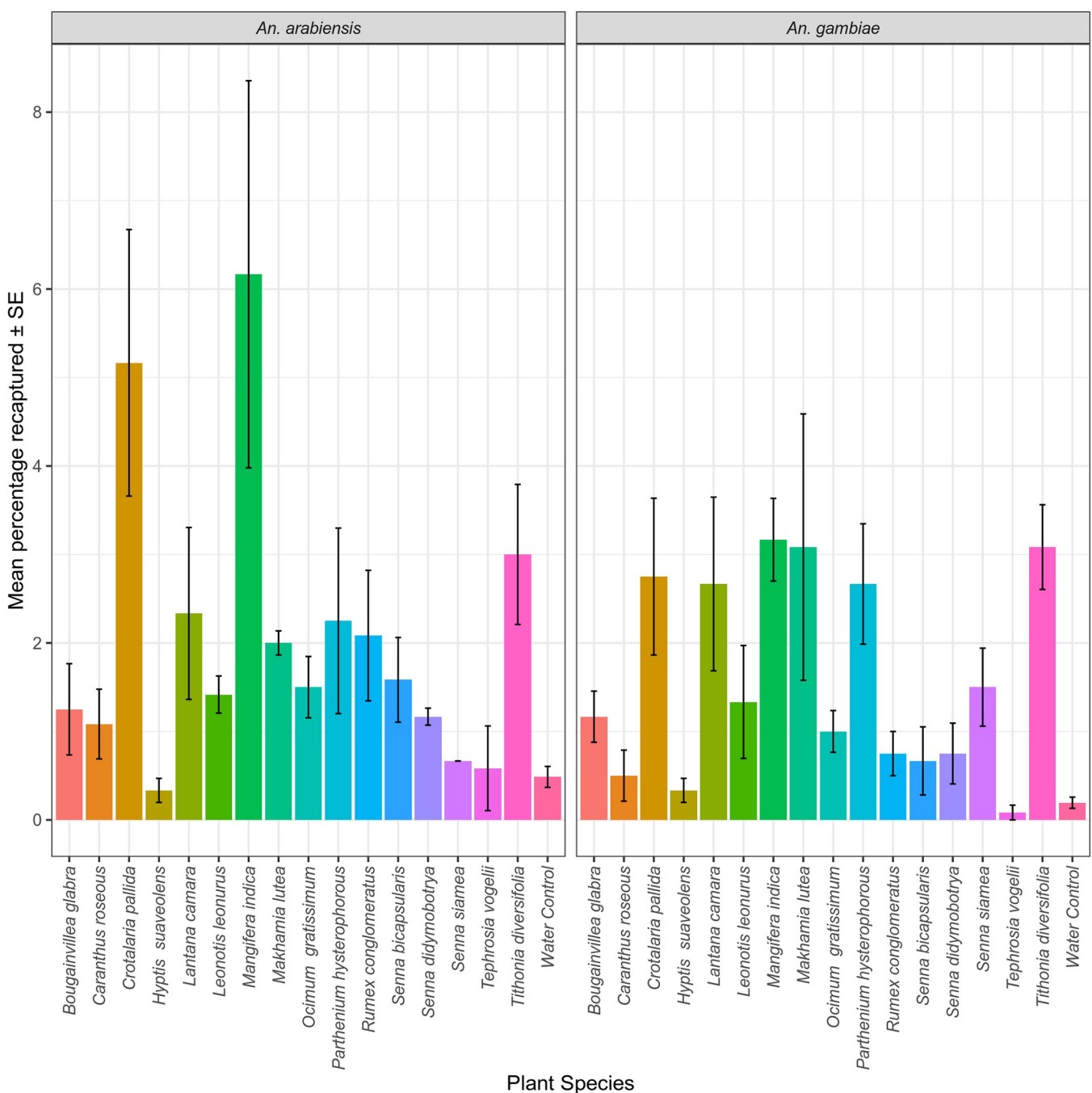

**Fig 2. The mean proportion of mosquitoes recaptured from 16 flowering plants in a multiple choice semi-field assay including a water control.** Error bars represent the standard error of the mean.

in these semi-field conditions (Fig 5). Finally, we compared the attractiveness of the newest ATSB v1.2 and the ATSB v1.1 against *Mangifera indica* in the fourth stage of the experiments.

## Comparison of *Mangifera indica* to ATSB v1.1 and 1.2

In experiment 4 comparing the most attractive flower *Mangifera indica*, with the ATSB versions 1.1 and 1.2 to *Anopheles* mosquitoes, the inclusion of test material variable led to a

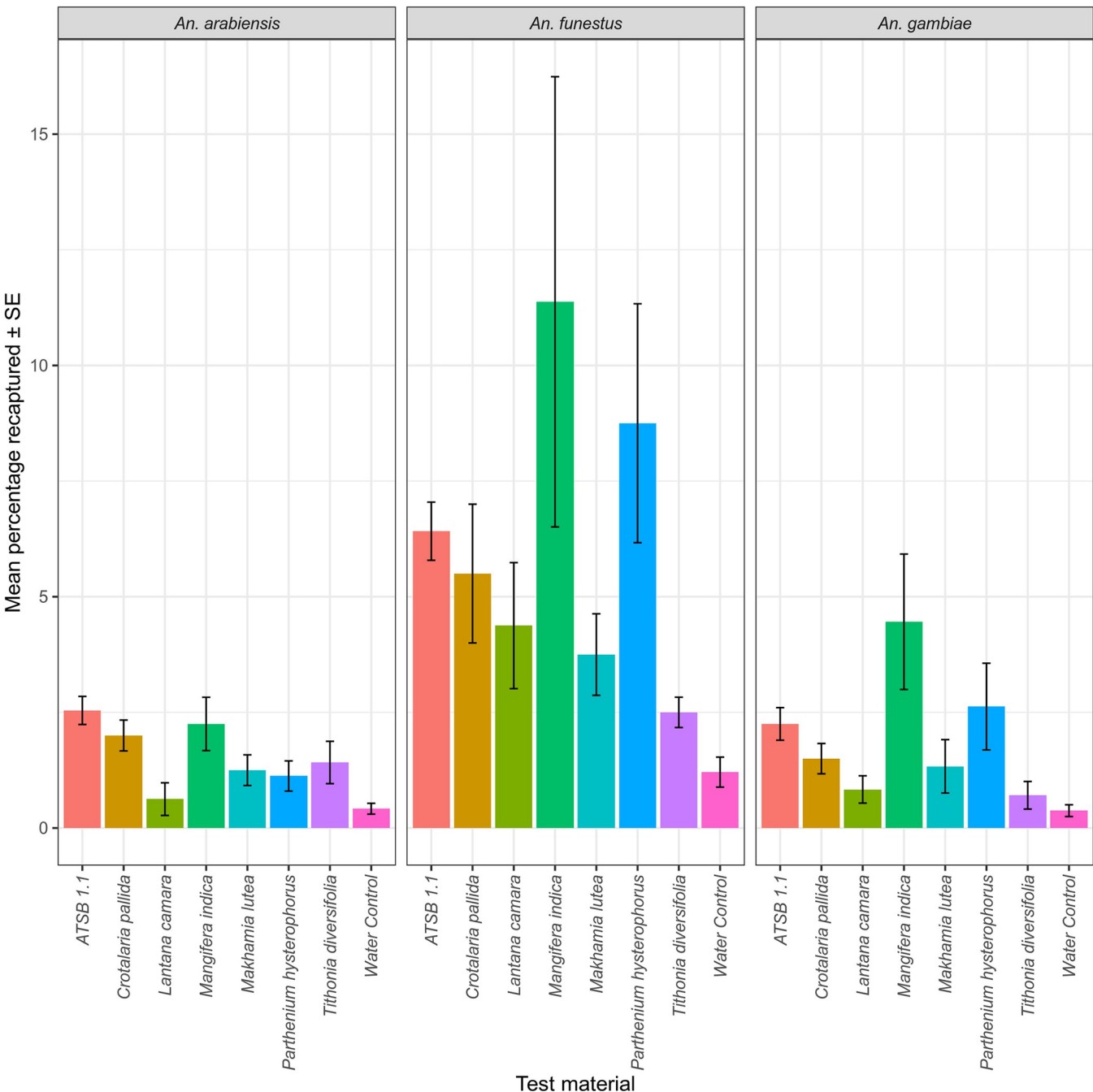

**Fig 3. The mean proportion of mosquitoes recaptured from the 6 most attractive flowering plants including ATSB v1.1, and water control in a multiple-choice semi-field assay.** Error bars represent the standard error of the mean.

significant improvement in the model fit for all three mosquito species compared to the null model. All three test materials (ATSB 1.1, ATSB 1.2 *Mangifera indica*) displayed a significant increase in their recapture rate compared to the control for all 3 Anopheles spp. The ATSB 1.2 significantly outperformed the ATSB 1.1 against *An. arabiensis*, *An. funestus*, and *An. gambiae*. The largest difference was seen in *An. funestus* where ATSB 1.2 (μ = 9.93%) displayed a greater attraction than the earlier iteration of ATSB 1.1 (μ = 6.03%; $Z = 5.54$, $p = <0.001$) and

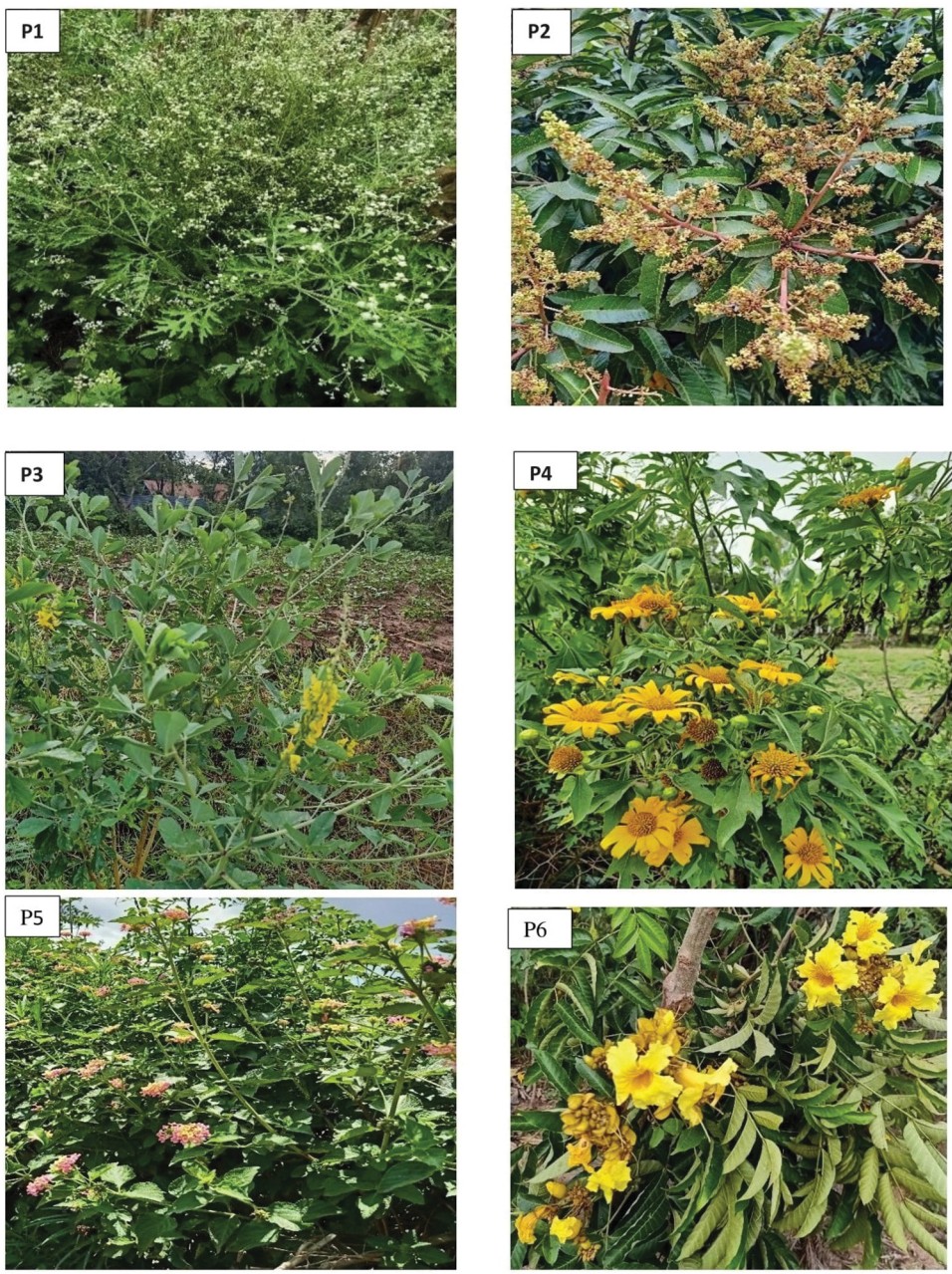

**Fig 4.** The most attractive flowering plants as found in experiment 1: *Parthenium hysterophorous* (P1), *Mangifera indica* (P2), *Crotalaria pallida* (P3), *Tithonia diversifolia* (P4), *Lantana camara* (P5) and *Makhamia lutea* (P6).

*Magnifera indica* (μ = 6.1%; $Z$ = 5.39, $p$ = <0.001). Similarly, the ATSB 1.2 (μ = 3.3%) slightly outcompeted the *Mangifera indica* for *An. arabiensis* (μ = 2.57%; $Z$ = 2.66, $p$ = 0.039) but had comparable recapture rate for *An. gambiae* (Fig 6).

## Discussion

This study established that different flowering plants in western Kenya have varying levels of attractancy to different local *Anopheles* species. The ATSB v1.2, which is the final version

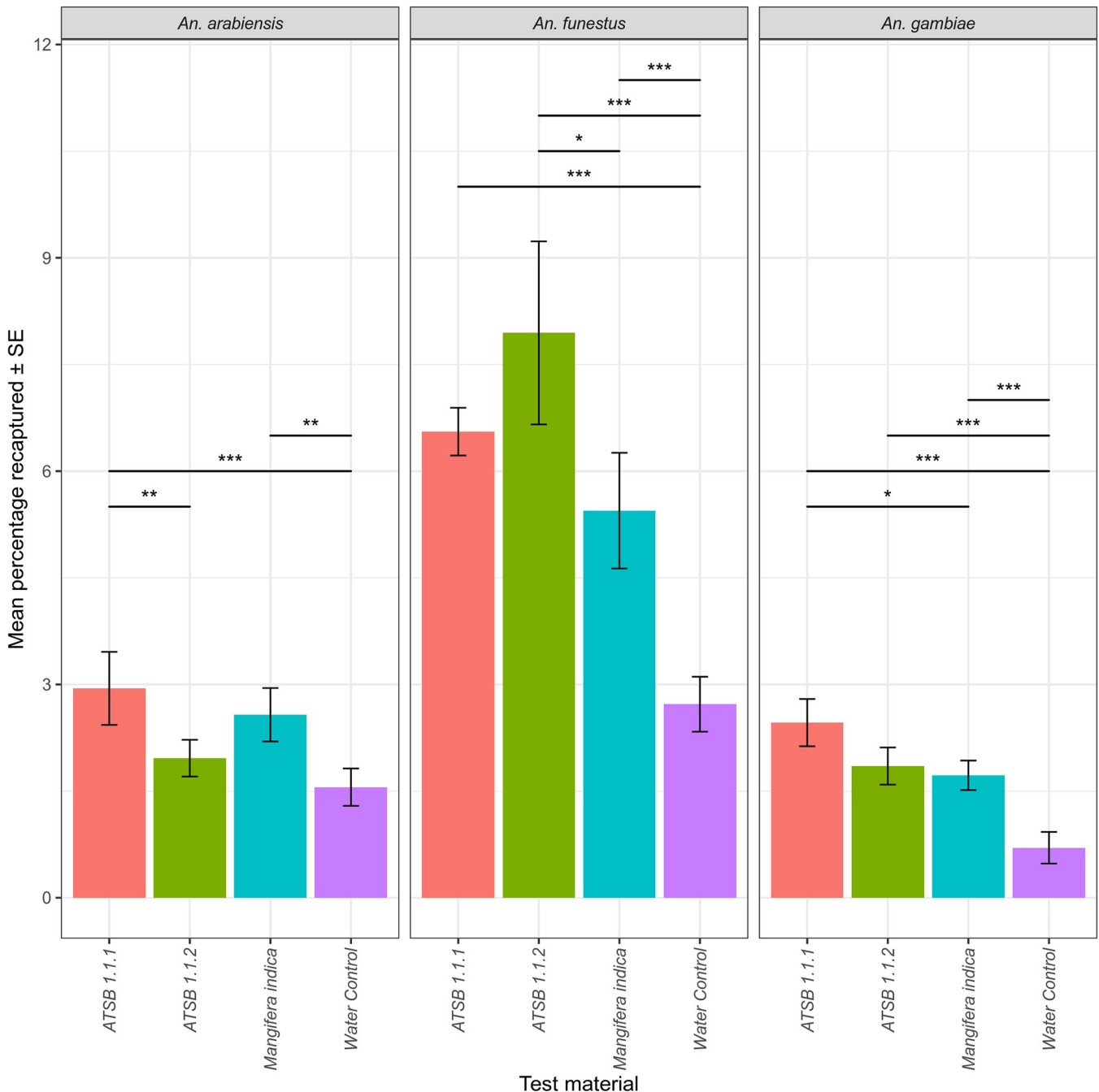

**Fig 5. The mean proportion of mosquitoes recaptured from the experiment comparing ATSB versions 1.1 and 1.1.2 with the most attractive plant;**
***Mangifera indica* and a water control.** P-value significance from a post-hoc Tukey's all-pair comparison indicated by '*' if ≤0.05, '**' if ≤0.01, and '***' if
≤0.001; non-significant relationships not shown.

currently undergoing epidemiological evaluation, was significantly more attractive to all the
three Anopheles species tested than the previous version 1.1. Additionally, the ATSB v1.2 was
significantly more attractive than the most attractive local flowering plant (*M. indica)*, tested to
*An. arabiensis, An. funestus* and a comparable mean percent recaptures with *An. gambiae.*
While this increase wasn't substantial it has established that the ATSBs can compete as

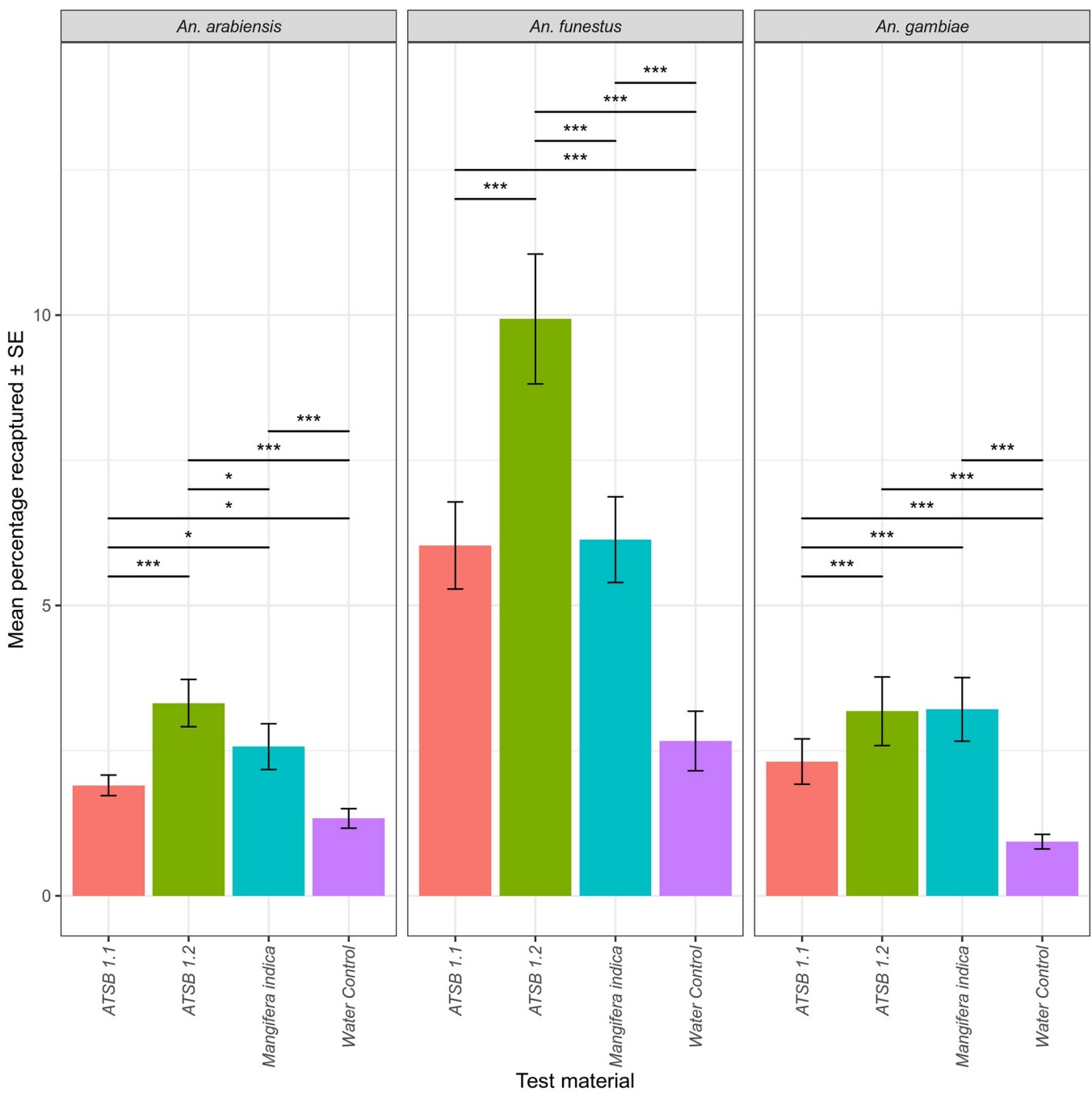

**Fig 6. The mean proportion of mosquitoes recaptured from the experiment comparing ATSB versions 1.1 and 1.2 with the most attractive plant;** *Mangifera indica* **and a water control.** P-value significance from a post-hoc Tukey's all-pair comparison indicated by '*' if ≤0.05, '**' if ≤0.01 and '***' if ≤0.001; non-significant relationships not shown.

potential sugar source expanding the evidence base for their potential successful use in non-arid environments.

The field experiment testing local malaria vector attraction to floral blossoms didn't yield the desired results after rigorous tests conducted at the study site. We attributed the failure to possibly a strong competition of volatiles released by the flora growing at the periphery of the

study site, given that in our study we used only branches of the flowering plants with a few flowers intact. Thus, the phytochemicals released by the many flowers on the remaining natural vegetation might have outperformed and diverted mosquitoes away from the glue-netted traps, which informed further analysis in the semi-field structures. Sugar feeding is a critical aspect of mosquito behavior, providing energy for flight, mate seeking, swarming, and egg laying [8]. The energy gained from sugar meals is necessary to drive dispersal and mosquito survivorship [30]. Accordingly, shrubs like the highly invasive *Prosopis juliflora* that serve as regular sugar sources to *An. gambiae* in Mali have been shown to accelerate the mosquito vectorial capacity [31] suggesting that clearing of such vegetation would reduce mosquito density by reducing nectar sources available.

Flowers typically are the most readily available sugar sources for mosquitoes though they occasionally may utilize seedpods, honeydew, and ripe-rotting fruits [10,32]. Mosquitoes rely on visual and olfactory cues to locate sugar resources in the environment [33,34]. Visual cues are chiefly responsible for a short-range attraction while odors play the role of long-range attraction to vectors [35]. Like other pollinator insects, mosquitoes in the field have been shown to be attracted to strongly fragrant and light-colored flowers [36,37] but only certain flower odors are attractive and can elicit mosquito aggregation and feeding [38]. In this study, a strong attraction was observed towards inflorescences of *M. indica*, *C. pallida*, *T. diversifolia*, *M. lutea*, and *P. hysterophorous*, implying that mosquito orientation to, and location of the flowers may have been due to odor released by the individual flowers. Previous studies in western Kenya indicated that *P. hysterophorous* is highly attractive to *An. gambiae* [39,40], and the *M. indica* flower was a preferred nectar source by male *An. gambiae* [14] which is consistent with our findings suggesting that these plants are beneficial to *Anopheles* vectors as sources of sugar.

Light colored flowers are frequently visited due to their ability to reflect light at night enabling their recognition by the nocturnal sugar seeking mosquitoes [41]. In our study, *M. lutea*, *C. pallida*, and *T. diversifolia* which have bright yellow flowers and the bright white inflorescence of *P. hysterophorous* were frequently selected while the dull *H. suaveolens* and *T. vogelii* were the least attractive, suggesting that flower color may be important in the selection of sugar sources by the *Anopheles* mosquitoes in this experiment despite them being enclosed in a mesh. Multimodal floral cues including odor and floral color, have been demonstrated to work together to initiate mosquito response to host plant volatiles [33]. Earlier studies identified components of the floral scents that are attractive to malaria vectors including terpenes, phenols, aliphatic esters, and aldehydes [11,42,43]. However, the current study did not isolate these phytochemical compounds from the tested flowers. Additional studies screening the attractive volatiles in these flowers may identify compounds for potential use in mosquito surveillance traps or to optimize future versions of ATSBs. Further, future studies should also explore the potential role of colour in attracting the vectors to ATSBs.

The ATSB was attractive to malaria mosquitoes of both sexes despite the availability of *M. indica* in semi-field conditions. This suggests that the attractant currently included in the ATSB may be effective in natural settings. Furthermore, the consistently higher attraction of *An. funestus* to the ATSB is a promising sign for the trial being conducted in *An. funestus* dominant area. However, given the consistently higher recapture rates in *An. funestus*, even in the plant species comparison, this may point to a potential species-specific difference that would need further investigation. This approach to mosquito control has been successful in reducing mosquito populations in dry areas [4,5,7,16,44] through a topical spray of ATSB on vegetation and using ATSB bait stations deployed on the outside walls of housing structures in Mali [18]. Although the different versions of ATSB had similarities in bait formulation,their attractiveness to the local malaria vectors varied. The ATSB v1.2 stood were superior to the previous

version 1.1 in the experiments. This disparity could have been caused by differences in the mode of production or membrane surfaces, which may have improved the bait solution retention. ATSB v1.2 was machine produced with a smooth membrane surface while v1.1 was manually produced with rough membranes prone to clogging with environmental dust. Additionally, a rapid depletion of the quantity of bait solution in the v1.1 membrane, might have led to the gradual reduction in the attractiveness of ATSB v1.1 compared to the v1.2 where the cases were rare. This is the first semi-field demonstration of the competitive advantage ATSBs have in luring mosquitoes when compared to natural sugar sources available in the local environment. The bait stations' extended efficacy period, technological simplicity, and oral mode of insecticide delivery as opposed to the conventional contact mode on LLINs and IRS, and ease of deployment make them a promising tool to manage residual malaria transmission [18,29,45]. Additionally, the bait station design limits the chances of contact between ATSB and non-target organisms as opposed to the spraying method used in previous studies [6,7,46]. Therefore, ATSBs present a potential option for the outdoor malaria vector control where impacts on non-target organisms is a concern.

The current study had a number of limitations. First, we may not have exhausted all the potential sugar sources as we only selected 16 flowering plants for our test based on their abundance in the natural environment from a botanical survey conducted in this area Yalla *et al.*, [Unpublished]". It is possible that some highly attractive flowers were out of their season and were missed by this study. However, if they were able to outcompete the ATSBs, it would only be seasonally. Furthermore, confining mosquitoes in semi-field conditions may have altered their behaviors towards the flowers or maybe the fact that the plants were confined in the glue-netted traps made them not truly visible at the normal range for the mosquitoes. Further, this experiment was conducted with a fixed quantity of flowering plants against a single ATSB in a controlled semi-field structure. Future work will need to investigate the role varying the density of both natural sugar sources and ATSBs have on attractiveness and how this may change seasonally. Finally, while the ATSBs showed some degrees of attraction compared to natural sugar sources in the semi-field experiments, it is unclear how they will perform outside the semi-field setting where a greater quantity of these attractive plants are available. Further studies on deployment may be required in different settings to optimize their effectiveness.

## Conclusion

Mosquitoes exhibit varying attractiveness to flowering plants in western Kenya. In a series of experiments, *M. indica* was the most attractive. The Westham ATSB v1.2 exhibited superior attractiveness in a semi-field setting indicating this product may be attractive to wild *Anopheles* mosquitoes despite the availability of natural sugar sources in the environment. This novel tool may prove suitable for use alongside existing vector control interventions but may require more field experiments to validate its attractiveness as well as an optimal deployment strategy.

## Supporting information

**S1 File. Attract_analysis_stats3and4_DATAset.**
(CSV)

**S2 File. Attract_perc_sum_fig1and2_DATAset.**
(CSV)

**S3 File. Attactancy_data.**
(XLSX)

**S4 File. Protocol_Attractancy study.**
(DOCX)

**S5 File. Semi-field structures & Trap locations.**
(DOCX)

## Acknowledgments

We thank the KEMRI-Entomology section staff for their dedicated support, which made it possible to draft this manuscript.

## Author Contributions

**Conceptualization:** Jackline Kosgei, Seline Omondi, Silas Agumba, Bernard Abong'o, John E. Gimnig, Angela F. Harris, Julian Entwistle, Peter R. Long, Eric Ochomo.

**Data curation:** Nick Yalla, Vincent Moshi.

**Formal analysis:** Nick Yalla, Daniel P. McDermott, Vincent Moshi, Eric Ochomo.

**Funding acquisition:** John E. Gimnig, Eric Ochomo.

**Investigation:** Nick Yalla, Brian Polo, Silas Agumba, John E. Gimnig, Peter R. Long, Eric Ochomo.

**Methodology:** Nick Yalla, Brian Polo, Jackline Kosgei, Seline Omondi, Silas Agumba, Vincent Moshi, Bernard Abong'o, John E. Gimnig, Peter R. Long, Eric Ochomo.

**Project administration:** Eric Ochomo.

**Supervision:** Brian Polo, Daniel P. McDermott, Jackline Kosgei, Seline Omondi, Bernard Abong'o, John E. Gimnig, Angela F. Harris, Julian Entwistle, Peter R. Long, Eric Ochomo.

**Validation:** Daniel P. McDermott, Vincent Moshi.

**Visualization:** Peter R. Long.

**Writing – original draft:** Nick Yalla, Brian Polo, Daniel P. McDermott.

**Writing – review & editing:** Jackline Kosgei, Seline Omondi, Silas Agumba, Vincent Moshi, Bernard Abong'o, John E. Gimnig, Angela F. Harris, Julian Entwistle, Peter R. Long, Eric Ochomo.

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
