## [Decision Letter · Decision Letter 0]

12 Dec 2022

PONE-D-22-29704A comparison of the attractiveness of flowering plant blossoms versus attractive targeted sugar baits (ATSBS) in western KenyaPLOS ONE

Dear Dr. Ochomo,

Thank you for submitting your manuscript to PLOS ONE. After careful consideration, we feel that it has merit but does not fully meet PLOS ONE’s publication criteria as it currently stands. Therefore, we invite you to submit a revised version of the manuscript that addresses the points raised during the review process.

We look forward to receiving your revised manuscript.

Kind regards,

George Dimopoulos, PhD MBA

Academic Editor

PLOS ONE

**Journal Requirements:**

5. We note that you have referenced (Yalla et al., unpublished) which has currently not yet been accepted for publication. Please remove this from your References and amend this to state in the body of your manuscript: (Yalla et al., [Unpublished]”) as detailed online in our guide for authors

6. We note that Figure 1 includes an image of a participant in the study.  

As per the PLOS ONE policy (http://journals.plos.org/plosone/s/submission-guidelines#loc-human-subjects-research) on papers that include identifying, or potentially identifying, information, the individual(s) or parent(s)/guardian(s) must be informed of the terms of the PLOS open-access (CC-BY) license and provide specific permission for publication of these details under the terms of this license. Please download the Consent Form for Publication in a PLOS Journal (http://journals.plos.org/plosone/s/file?id=8ce6/plos-consent-form-english.pdf). The signed consent form should not be submitted with the manuscript, but should be securely filed in the individual's case notes. 

Please amend the methods section and ethics statement of the manuscript to explicitly state that the participant has provided consent for publication: “The individual in this manuscript has given written informed consent (as outlined in PLOS consent form) to publish these case details”. 

Reviewers' comments:

Reviewer's Responses to Questions

**Comments to the Author**

1. Is the manuscript technically sound, and do the data support the conclusions?

Reviewer #1: Partly

Reviewer #2: Partly

2. Has the statistical analysis been performed appropriately and rigorously? 

Reviewer #1: No

Reviewer #2: I Don't Know

3. Have the authors made all data underlying the findings in their manuscript fully available?

Reviewer #1: Yes

Reviewer #2: No

4. Is the manuscript presented in an intelligible fashion and written in standard English?

Reviewer #1: Yes

Reviewer #2: No

5. Review Comments to the Author

Reviewer #1: In their manuscript ‘A comparison of the attractiveness of flowering plant blossoms versus attractive targeted sugar baits (ATSBS) in western Kenya’, the authors examine whether ATSBs are more attractive to Anopheles mosquitoes than 16 local flowering plants in a semi-field environment in Kenya. They use a release and recapture design based involving three different anopheline species: An gambiae, An arabiensis, and An funestus, and compare the attractiveness of different plants and ATSBs based on raw re-capture counts. They identify 6 different plants as being highly attractive to mosquitoes, with Mangifera indica determined to be the most attractive. They then compare the attractiveness of these plants to three different ATSB formulations. For two ATSBs, they find greater attractiveness than water but no clear difference to M. indica. For the final ATSB, they find that it is more attractive than M. indica. Given the increasing relevance of ATSBs to mosquito control, the research question is highly relevant, however, I have some issues with the experimental design and the approach to data analysis.

Major comments:

1. There is potentially a major and systemic issue with the was that the data have been analyzed. Analyses are based on mean number of mosquitoes collected. However, it appears as though the number of mosquitoes released varied depending on the mosquito species and the experiment being performed. Comparing across these treatments becomes problematic when some treatments have a high N and others have a low N, and this issue could potentially lead to drastic differences in conclusions drawn from some experiments, particularly if there was replicate-by-replicate variation in samples size. My recommendation is to re-analyze the data using percentage recaptured values rather than raw/absolute recapture values. Alternatively, you could demonstrate that there is no difference in outcome between the two data formats.

2. One thing that is missing from the experimental design and data in this study is data from no choice assays. i.e., where mosquitoes are released into a cage and presented with only an ATSB or only a flowering plant. Such an experiment would facilitate estimates of the rate of change in attractiveness of any bait station when other attractants are present and would provide important supporting evidence to support your conclusion that ATSB-containing bait stations are still effective when mosquitoes have a choice of attractants.

3. The following information is missing from the methods section:

3a: Lines 85-87 – “Prior to the semi-field experiments, a team of entomologists and botanists conducted a botanical survey in Asembo (-0.1837oS, 34o23′1‵‵ E), Siaya County to characterize the common flora of the area.” How extensive was this survey? How much time was spent? What area was covered? What time of year was the survey conducted? More detail in the text would be useful.

3b: More details on plant collection are required. Where were the plants used in semi-field assays sampled from? Did this encompass multiple sites? The text should specify whether there was consistency in the plant material that was used between experiments.

3c: What procedures were followed to make certain that the field cage was cleared of mosquitoes between experiments? Specify in the text.

3d: The methods section does not adequately describe replication for each of the 4 semi-field experiments. i.e., how many bait stations per treatment per experiment. It’s important for your readers to know how many times each plant/ATSB was tested with each mosquito species.

3e: No information on the composition of the three ATSB formulations is provided. I recognize that this is likely proprietary knowledge, but without some idea about the inherent differences between ATSBs v1.1.1, v1.1.2, and v1.2, the relevance of your results is diminished. Surely there is scope to discuss variation in types of ingredients/components? i.e., a different adjuvant, a different attractant, increased concentration of attractant etc.

4. The entire results section should be adjusted so that key findings of experiments are presented in the context of output from your statistical output. As each subsequent experiment in your study is informed by the results of a previous experiment, it is necessary to demonstrate to your readers why you have made the decisions you have made. i.e., why you selected those six specific plants from experiment 1 for further analysis in experiment 2. At the moment, the text does not have that level of rigor.

Minor comments:

5. Missing information and issues with figures and figure legends:

• Figure 2 could note the 6 most attractive plants selected for further testing.

• There are no x-axis titles for any figs

• Fig 3 legend – “captured from the 6 most attractive flowering plants” – this is not explained well

• Figs 5/6 should specify the statistical groups for the 4 data sets in each panel.

6. Line 52 – ‘source’ change to ‘sources’

7. The sentence starting on line 99 is missing commas.

8. Lines 106 and 149. “3-5 days old” this is inaccurate. Change to 3-5 days post-eclosion or 3-5 days post-emergence.

9. Lines 170-171 – “The six test flowers were randomized for the 3 semi-field structures” – the meaning of this sentence is unclear and this should be re-phrased and explained in greater detail.

10. Lines 181-185 – “These ATSB versions are similar in bait formulation although v1.1.1 and v1.1.2 were manually produced contained a rain proof layer while v1.2 is machine-produced with a rain proof layer, retention of more quantity of bait solution and more volatile release for more than 6 months due to excellent sealing between the trays and bait station membrane.” - This sentence does not make sense.

11. Lines 231-232 – “Other flowering plants found to be attractive to the malaria vectors were P. hysterophorous, C. pallida, M. lutea, T. diversifolia and L. camara (Figure 4).” – this text adds nothing. It’s just a repeat of test from lines 222-223.

12. Table 2 title – materials spelled incorrectly

13. Table 2 legend – P values - > used in place of <

14. Line 267 - “different flowering plants” – the text should specify that this is in a region of Kenya and that they attract local vector species.

15. Lines 311-314 – “The bait stations’ extended efficacy period, technological simplicity, and oral mode of insecticide delivery as opposed to the conventional contact mode on LLINs and IRS, and ease of deployment make them a promising tool to manage residual malaria transmission.” References are needed for this statement as many parameters are not examined in this study.

Reviewer #2: Lane 35 & 36. Based on the data results ATSB v1.2 is significantly more attractive compared to Mangifera indica and non-significantly attractive to ATSB v1.1. Please check.

Lane 98. Please mention the mosquito species identification method after mosquito collection in methodology section.

Lane 117. Please write about the details of ‘open field’

Lane 133. How many mosquitoes were released in this experiment and mention the release time and species names?

Lane 147. Please mention the ‘F1’ progeny of An. funestus adult mosquitoes were released.

Lane 175. Please describe about “Nine sets of test materials were evaluated”.

Lane 175-177. Looks like a reptation of lane 173-175.

Lane 180 & 181. Please abbreviate ATSB versions at their first use and follow the same in entire the MS. Correct everywhere carefully to avoid the misunderstandings of different ATSB versions.

Lane 186. Explain briefly about active ingredient “Dinotefuran” and its effect on non-target organisms and its approvals to use in public health in Intro section.

Lane 190. Please provide the full picture of semi-field and show corners where traps were placed.

Lane 210. Mention details of other mosquito species and insects collected other than released ones in the traps.

Lane 215. Describe Experiment 1, 2, 3, & 4 and also mention in methodology section (Field and semi-field experiments)

Lane 229: “ATSB v1.1.1 and water control” is confusing the sentence.

Lane 244. “ATSB version 1.1.2 did not show any greater attraction than the water control” Describe interpretation in discussion section.

Lane 243. “An. arabiensis (p>0.05)” is not match with Table 2 data,

Lane 245 & 246: Cross check with Lane 35 & 36.

Lane 248-250: Check and correct the P-values.

Lane 334. Author should provide a possible reason for ATSB version 1.2 more attracted compared to v1.1 as they mention it differ only in outer membrane with similar bait formulation.

6. PLOS authors have the option to publish the peer review history of their article (what does this mean?). If published, this will include your full peer review and any attached files.

Reviewer #1: No

Reviewer #2: No

---

## [Author Response · Author response to Decision Letter 0]

7 Feb 2023

Comments to the Author

1. Is the manuscript technically sound, and do the data support the conclusions?

Reviewer #1: Partly

Reviewer #2: Partly

2. Has the statistical analysis been performed appropriately and rigorously?

Reviewer #1: No

Reviewer #2: I Don't Know

3. Have the authors made all data underlying the findings in their manuscript fully available?

Reviewer #1: Yes

Reviewer #2: No

4. Is the manuscript presented in an intelligible fashion and written in standard English?

Reviewer #1: Yes

Reviewer #2: No

5. Review Comments to the Author

Reviewer #1: In their manuscript ‘A comparison of the attractiveness of flowering plant blossoms versus attractive targeted sugar baits (ATSBS) in western Kenya’, the authors examine whether ATSBs are more attractive to Anopheles mosquitoes than 16 local flowering plants in a semi-field environment in Kenya. They use a release and recapture design based involving three different anopheline species: An gambiae, An arabiensis, and An funestus, and compare the attractiveness of different plants and ATSBs based on raw re-capture counts. They identify 6 different plants as being highly attractive to mosquitoes, with Mangifera indica determined to be the most attractive. They then compare the attractiveness of these plants to three different ATSB formulations. For two ATSBs, they find greater attractiveness than water but no clear difference to M. indica. For the final ATSB, they find that it is more attractive than M. indica. Given the increasing relevance of ATSBs to mosquito control, the research question is highly relevant, however, I have some issues with the experimental design and the approach to data analysis.

Major comments:

1. There is potentially a major and systemic issue with the was that the data have been analyzed. Analyses are based on mean number of mosquitoes collected. However, it appears as though the number of mosquitoes released varied depending on the mosquito species and the experiment being performed. Comparing across these treatments becomes problematic when some treatments have a high N and others have a low N, and this issue could potentially lead to drastic differences in conclusions drawn from some experiments, particularly if there was replicate-by-replicate variation in samples size. My recommendation is to re-analyze the data using percentage recaptured values rather than raw/absolute recapture values. Alternatively, you could demonstrate that there is no difference in outcome between the two data formats.

Response: We appreciate this comment,the data was re-analysed using percentage recapture as suggested

2. One thing that is missing from the experimental design and data in this study is data from no-choice assays. i.e., where mosquitoes are released into a cage and presented with only an ATSB or only a flowering plant. Such an experiment would facilitate estimates of the rate of change in the attractiveness of any bait station when other attractants are present and would provide important supporting evidence to support your conclusion that ATSB-containing bait stations are still effective when mosquitoes have a choice of attractants. 

Response: The phenomenon of attractancy is quite difficult to evaluate in a cage. The attractancy experiment whose results are reported in this study first compared flower blossoms attractiveness to mosquitoes. From this we learnt that flowers have varying levels of attractiveness to local malaria vectors. Based on previous studies and durability assays conducted with the bait stations in Israel and Mali, we understand that ATSB has a competitive ability against natural sugar sources (These papers from Mali and Israel are cited in this manuscript). Therefore, our study relied on these established facts by previous studies.

3. The following information is missing from the methods section:

3a: Lines 85-87 – “Prior to the semi-field experiments, a team of entomologists and botanists conducted a botanical survey in Asembo (-0.1837oS, 34o23′1‵‵ E), Siaya County to characterize the common flora of the area.” How extensive was this survey? How much time was spent? What area was covered? What time of year was the survey conducted? More detail in the text would be useful. Response: We added more details that read “The survey estimated the cover of all flowering plants species around the peridomestic spaces across ten clusters between May-August 2020. These clusters had wide geographical coverage made up of groups of adjacent villages with households ranging between 100-400 on average.”

3b: More details on plant collection are required. Where were the plants used in semi-field assays sampled from? Did this encompass multiple sites? The text should specify whether there was consistency in the plant material that was used between experiments. 

Response: The same flowering plants tested in the field are the same flowers that were tested in the semi-field in this study. Western Kenya has a homogeneous pattern of vegetation distribution. Therefore, most floral species in Asembo are also available around the research station except for a few that were out of season and were sourced daily freshly from Asembo and transported for testing at KEMRI in the semi-field structures. This information is now documented in the manuscript. Lines 85-88

3c: What procedures were followed to make certain that the field cage was cleared of mosquitoes between experiments? Specify in the text. 

Response: This information was inserted in the manuscript. At least two days time gap was left in between the experiments for any remaining mosquito to die through natural mortality. Lines 162-164

3d: The methods section does not adequately describe replication for each of the 4 semi-field experiments. i.e., how many bait stations per treatment per experiment. It’s important for your readers to know how many times each plant/ATSB was tested with each mosquito species. 

Response: This information is captured well between lines 172-189. In brief, each semi-field could accommodate 4 traps, so; in experiment 1, each time three flowers vs water control was tested. In experiment 2, one ATSB V1.1, two flowers and water control. Experiment three, we tested two ATSB versions (v1.1 & v1.1.2), most attractive flower and water control. Finally, in experiment four, we tested the most attractive flower, ATSB v1.1 & v 1.2, and water control. These are well documented in the methods section.

3e: No information on the composition of the three ATSB formulations is provided. I recognize that this is likely proprietary knowledge, but without some idea about the inherent differences between ATSBs v1.1.1, v1.1.2, and v1.2, the relevance of your results is diminished. Surely there is scope to discuss variation in types of ingredients/components? i.e., a different adjuvant, a different attractant, increased concentration of attractant etc. 

Response: The ATSB produced by Westham, and tested in this study were versions 1.1, 1.1.2, and 1.2. Version which is currently under epidemiological trial in Kenya, Zambia, and Mali. The ATSB versions were similar in bait formulation. These ATSBs contained an attractant, dinotefuran as the active ingredient, which is an oral neonicotinoid insecticide, sugar as a feeding stimulant and Bittrex added to deter human consumption. The dinotefuran insecticide has been indicated to be of low toxicity to humans and has a negligible effect on non-target organisms. These statements are cited as well in the manuscript lines 190-197.

4. The entire results section should be adjusted so that key findings of experiments are presented in the context of output from your statistical output. As each subsequent experiment in your study is informed by the results of a previous experiment, it is necessary to demonstrate to your readers why you have made the decisions you have made. i.e., why you selected those six specific plants from experiment 1 for further analysis in experiment 2. At the moment, the text does not have that level of rigor. 

Response: The adjustments have now been made to indicate what next was done after each stage of the experiments in the study. See the result section.

Minor comments:

5. Missing information and issues with figures and figure legends:

• Figure 2 could note the 6 most attractive plants selected for further testing. 

Response: Edited as suggested in the manuscript.

• There are no x-axis titles for any figs. 

Response: This was an oversight and has since been corrected

• Fig 3 legend – “captured from the 6 most attractive flowering plants” – this is not explained well Response: This was adjusted by re-analysis of the data. See the result section.

• Figs 5/6 should specify the statistical groups for the 4 data sets in each panel. 

Response: These were added after conducting a re-analysis of the data. See figure legends

6. Line 52 – ‘source’ change to ‘sources’ 

Response: this was changed.

7. The sentence starting on line 99 is missing commas. 

Response: The sentence was punctuated 

8. Lines 106 and 149. “3-5 days old” this is inaccurate. Change to 3-5 days post-eclosion or 3-5 days post-emergence. 

Response: This statement was adjusted to post-emergence

9. Lines 170-171 – “The six test flowers were randomized for the 3 semi-field structures” – the meaning of this sentence is unclear and this should be re-phrased and explained in greater detail. Response: Randomization was to enable us know which of the two flowers would be tested against the other since we had 3 semi-field structures of which one could allow testing of 2 flowers, 1 ATSB and water control at this stage of the experiment. The sentence was re-phrased.

10. Lines 181-185 – “These ATSB versions are similar in bait formulation although v1.1.1 and v1.1.2 were manually produced contained a rain proof layer while v1.2 is machine-produced with a rain proof layer, retention of more quantity of bait solution and more volatile release for more than 6 months due to excellent sealing between the trays and bait station membrane.” - This sentence does not make sense. 

Response: This section has been revised extensively for clarity

11. Lines 231-232 – “Other flowering plants found to be attractive to the malaria vectors were P. hysterophorous, C. pallida, M. lutea, T. diversifolia and L. camara (Figure 4).” – this text adds nothing. It’s just a repeat of test from lines 222-223. 

Response: This line was deleted

12. Table 2 title – materials spelled incorrectly- 

Response: This table was deleted after re-analysis as we found it to be unnecessary.

13. Table 2 legend – P values - > used in place of <

Response: Now corrected by reanalysis; the table was deleted

14. Line 267 - “different flowering plants” – the text should specify that this is in a region of Kenya and that they attract local vector species. 

Response: This information was inserted to reflect western Kenyan region in the discussion section.

15. Lines 311-314 – “The bait stations’ extended efficacy period, technological simplicity, and oral mode of insecticide delivery as opposed to the conventional contact mode on LLINs and IRS, and ease of deployment make them a promising tool to manage residual malaria transmission.” References are needed for this statement as many parameters are not examined in this study. Response: Additional citations/ references demonstrating these qualities from previous studies with the bait station were added.

Reviewer #2:

 Lane 35 & 36. Based on the data results ATSB v1.2 is significantly more attractive compared to Mangifera indica and non-significantly attractive to ATSB v1.1. Please check. 

Response: After re-analysis of the data, this error was eliminated; we found that the two versions of ATSB were in most cases significantly attractive compared to the most attractive Mangifera indica although the ATSB v1.2, currently under epidemiological trial was superior to the most attractive flower or the older ATSB versions. See result section

Lane 98. Please mention the mosquito species identification method after mosquito collection in methodology section. 

Response: The mosquito species identification was done morphologically, based on the information from the previous studies conducted in these sites. This information was added in the manuscript, see lines 119 and 120 for the reference studies.

Lane 117. Please write about the details of ‘open field’ 

Response: more information was added concerning the field site, Lines 120-126

Lane 133. How many mosquitoes were released in this experiment and mention the release time and species names? 

Response: n the field experiment we did not release any mosquito as we expected wild mosquitoes to be attracted to our traps. However, after a 20-day experiment we only managed to collect 132 culicine mosquitoes on the traps and negligible anopheles mosquito catches. This is the reason why we shifted the experiment to semi-field where conditions were more controlled than in open field. We attributed the failure to collect Anopheles mosquitoes to the potential competition from volatiles of the naturally rooted flowering plants that overpowered the volatiles from the test flowers and subsequently diverted mosquitoes from the traps.

Lane 147. Please mention the ‘F1’ progeny of An. funestus adult mosquitoes were released. 

Response: This was inserted in the manuscript.

Lane 175. Please describe about “Nine sets of test materials were evaluated”. 

Response: This was supposed to be 9 replicates of test materials, now corrected in the manuscript.

Lane 175-177. Looks like a repetition of lane 173-175. 

Response: The statement was edited.

Lane 180 & 181. Please abbreviate ATSB versions at their first use and follow the same in entire the MS. Correct everywhere carefully to avoid the misunderstandings of different ATSB versions. 

Response: The ATSB versions were abbreviated appropriately as suggested.

Lane 186. Explain briefly about active ingredient “Dinotefuran” and its effect on non-target organisms and its approvals to use in public health in Intro section. 

Response: This information was explained and referenced. See lines 190-197 for the explanation and references. Briefly, Dinotefuran is an oral insecticide from the class Neonecotinoid that has been proven to pose minimal health risk to human and non-target organisms. We included citations to the descriptions in the manuscript.

Lane 190. Please provide the full picture of semi-field and show corners where traps were placed. Response: The semi-field pictures and the four corners where traps are set will be provided as a supplementary document; highlighting all the four sides.

Lane 210. Mention details of other mosquito species and insects collected other than released ones in the traps. 

Response: There was no other mosquito species collected other than that which was released for each experiment.

Lane 215. Describe Experiment 1, 2, 3, & 4 and also mention in methodology section (Field and semi-field experiments). 

Response: The experimental stages 1, 2, 3, & 4 are well explained under the semi-field experiment section. Experiment 1 compared the relative attractiveness of the 16 flowering plants. Experiment 2 compared ATSB v1.1.1 and the 6 most attractive flowers. The experiment 3 compared the most attractive flower, Mangifera indica vs two versions of ATSB 1.1.1 & v1.1.2. Finally, experiment 4 compared the ATSB v1.1 & v1.2 with the most attractive flowers. See lines 170-191 for the descriptions.

Lane 229: “ATSB v1.1.1 and water control” is confusing the sentence. 

Response: The statement was edited to read,” the second experiment evaluating the six most attractive flower species in comparison to ATSB v1.1 and water control found that Mangifera indica had the highest mosquito collection and was the most preferred flower by both sexes of the three mosquito species.

Lane 244. “ATSB version 1.1.2 did not show any greater attraction than the water control” Describe interpretation in discussion section. 

Response: This error was corrected by re-analysis of the data; see result section

Lane 243. “An. arabiensis (p>0.05)” is not match with Table 2 data, 

Response: The table was removed after re-analysis of the data which also eliminated this error henceforth.

Lane 245 & 246: Cross check with Lane 35 & 36. 

Response: Noted and corrected; see manuscript

Lane 248-250: Check and correct the P-values. 

Response: The table was removed after re-analysis of the data.

Lane 334. Author should provide a possible reason for ATSB version 1.2 more attracted compared to v1.1 as they mention it differ only in outer membrane with similar bait formulation.

Response: The version 1.1 was produced several months earlier than v1.2, and had features like rough membrane surface and being manually produced. The rough membrane would trap dust which perhaps clogged the pores releasing volatiles, hence with time lowered its attractiveness. These features could be responsible for the reduction in v1.1 attractiveness with time. On the other hand, v1.2 had smooth membrane surface, machine produced, better bait solution retention, sustained slow-release of the volatiles and excellent sealing between the bait station tray and the membrane. These features could have been the reasons for its superiority in attracting mosquitoes over the previous versions.

---

## [Decision Letter · Decision Letter 1]

6 Apr 2023

PONE-D-22-29704R1A comparison of the attractiveness of flowering plant blossoms versus attractive targeted sugar baits (ATSBS) in western KenyaPLOS ONE

Dear Dr. Ochomo,

Thank you for submitting your manuscript to PLOS ONE. After careful consideration, we feel that it has merit but does not fully meet PLOS ONE’s publication criteria as it currently stands. Therefore, we invite you to submit a revised version of the manuscript that addresses the points raised during the review process, that involves some editing of the manuscript text as suggested by one of the reviewers.  

We look forward to receiving your revised manuscript.

Kind regards,

George Dimopoulos, PhD MBA

Academic Editor

PLOS ONE

Journal Requirements:

Reviewers' comments:

Reviewer's Responses to Questions

**Comments to the Author**

1. If the authors have adequately addressed your comments raised in a previous round of review and you feel that this manuscript is now acceptable for publication, you may indicate that here to bypass the “Comments to the Author” section, enter your conflict of interest statement in the “Confidential to Editor” section, and submit your "Accept" recommendation.

Reviewer #1: All comments have been addressed

Reviewer #2: (No Response)

2. Is the manuscript technically sound, and do the data support the conclusions?

Reviewer #1: Yes

Reviewer #2: Partly

3. Has the statistical analysis been performed appropriately and rigorously? 

Reviewer #1: Yes

Reviewer #2: I Don't Know

4. Have the authors made all data underlying the findings in their manuscript fully available?

Reviewer #1: Yes

Reviewer #2: Yes

5. Is the manuscript presented in an intelligible fashion and written in standard English?

Reviewer #1: Yes

Reviewer #2: Yes

6. Review Comments to the Author

Reviewer #1: (No Response)

Reviewer #2: Major comments

Please describe all versions of ATSBS under subheading in “Materials and Methods” section.

Figure 5 showed significant result against An. arabiensis in all test groups, however, results section showed non-significant p-values in Lane 251 and 253. Please check and correct the figure.

Lane no 266: Mentioned “non-significant relationships not shown” however results showed (lane 253 and 255) non-significant ‘p’ values.

Minor comments

Remove the underline symbol below degree Celsius (27-30 ºC)

I find different terminology used in MS for flowering plants. I suggest single terminology would be better.

Correct the typo error in “An. Arabiensis”

Lane 271: Current the typo error “ATSB 1.2 Mangifera indica”

7. PLOS authors have the option to publish the peer review history of their article (what does this mean?). If published, this will include your full peer review and any attached files.

Reviewer #1: No

Reviewer #2: No

---

## [Author Response · Author response to Decision Letter 1]

27 Apr 2023

Response to Reviewers.

Journal Requirements:

Response: The reference list is complete and correct based on my review. No retracted paper was cited. Therefore, no changes were made to the list.

Reviewer #2: Major comments

Please describe all versions of ATSBS under subheading in “Materials and Methods” section.

Response-We’ve documented the notable differences and similarities in the three versions of the ATSB tested. Again, via personal information from the manufacturer, we learnt that despite the similarity in bait formulation, there were a few differences noted in ( v1.1 & 1.1.2) that ultimately led to the manufacture of the final version 1.2 used in the main trial. These differences are in the membrane texture, bait retention ability, and longevity but this is information we are only reporting based on what is disclosed to us by the manufacturer

Figure 5 showed significant result against An. arabiensis in all test groups, however, results section showed non-significant p-values in Lane 251 and 253. Please check and correct the figure. 

Response-The error in figure five was fixed. See attached figure 5.

Lane no 266: Mentioned “non-significant relationships not shown” however results showed (lane 253 and 255) non-significant ‘p’ values.

Response-This was corrected after fixing figure 5

Minor comments

Remove the underline symbol below degree Celsius (27-30 ºC)

Response – this was corrected

I find different terminology used in MS for flowering plants. I suggest single terminology would be better.

Response - The flowering part of a plant that will form the seeds or fruit is called a blossom and synonyms include bloom and flower. It really doesn’t alter the meaning of a flower.

Correct the typo error in “An. Arabiensis”

Response: The typo was corrected

Lane 271: Current the typo error “ATSB 1.2 Mangifera indica”

Response - In experiment 3 the ATSB version tested was labeled v1.1.2 which differs in labeling to the final ATSB version 1.2. The two versions are labeled so in the MS.

---

## [Editor Report · Decision Letter 2]

22 May 2023

A comparison of the attractiveness of flowering plant blossoms versus attractive targeted sugar baits (ATSBS) in western Kenya

PONE-D-22-29704R2

Dear Dr. Ochomo,

We’re pleased to inform you that your manuscript has been judged scientifically suitable for publication and will be formally accepted for publication once it meets all outstanding technical requirements.

Kind regards,

George Dimopoulos, PhD MBA

Academic Editor

PLOS ONE
---

## [Editor Report · Acceptance letter]

29 May 2023

PONE-D-22-29704R2 

A comparison of the attractiveness of flowering plant blossoms versus attractive targeted sugar baits (ATSBs) in western Kenya. 

Dear Dr. Ochomo:

I'm pleased to inform you that your manuscript has been deemed suitable for publication in PLOS ONE. Congratulations! Your manuscript is now with our production department. 

Kind regards, 

on behalf of

Dr. George Dimopoulos 

Academic Editor

PLOS ONE